# Identification of Pinocembrin as an Anti-Glycation Agent and α-Glucosidase Inhibitor from Fingerroot (*Boesenbergia rotunda*): The Tentative Structure–Activity Relationship towards MG-Trapping Activity

**DOI:** 10.3390/molecules23123365

**Published:** 2018-12-19

**Authors:** Thammatee Potipiranun, Sirichai Adisakwattana, Wisuttaya Worawalai, Rico Ramadhan, Preecha Phuwapraisirisan

**Affiliations:** 1Program of Biotechnology, Faculty of Science, Chulalongkorn University, Bangkok 10330, Thailand; yoku62@hotmail.com; 2Center of Excellence in Natural Products, Department of Chemistry, Faculty of Science, Chulalongkorn University, Bangkok 10330, Thailand; yok_cu@hotmail.com (W.W.); lingkungan_krus@yahoo.co.id (R.R.); 3Department of Nutrition and Dietetics, Faculty of Allied Health Sciences, Chulalongkorn University, Bangkok 10330, Thailand; adisakwattana_siri@yahoo.com; 4Department of Chemistry, Faculty of Science and Technology, Airlangga University, Surabaya 60115, Indonesia

**Keywords:** α-glucosidase, *Boesenbergia rotunda*, advanced glycation end-products, methylglyoxal, pinocembrin, structure-activity relationship, diabetes

## Abstract

Diabetes mellitus (DM) is a disease that is caused by a malfunction of carbohydrate metabolism, which plays an important role in the development of long-term diabetic complications. The excess glucose can be transformed to methylglyoxal (MG), a potential precursor of glycation. Glycation is a spontaneous non-enzymatic reaction that initially yields advanced glycation end-products (AGEs), which ultimately triggers several severe complications. Therefore, the inhibition of AGEs formation is the imperative approach for alleviating diabetic complications. The aim of this research was to investigate the glycation and α-glucosidase inhibitory abilities of compounds isolated from fingerroot. The dichloromethane extract afforded three flavanones, two chalcones, two dihydrochalcones, and one kavalactone. Most of the isolated compounds showed higher inhibition effect against AGEs formation than aminoguanidine (AG). Subsequent evaluation in MG-trapping assay indicated that their trapping potency was relatively comparable to AG. Their structure-activity relationships (SAR) of MG-trapping activity were investigated using the comparison of the structures of flavonoids. In addition, pinocembrin displayed moderate α-glucosidase inhibition against both maltase and sucrose, with IC_50_ values of 0.35 ± 0.021 and 0.39 ± 0.020 mM, respectively.

## 1. Introduction

Diabetes mellitus is a metabolic disease caused by a malfunction of carbohydrate metabolism characterized by high blood glucose level (hyperglycemia). Chronic hyperglycemia in diabetes results in the development of long-term diabetic complication, and associates it with the increased risk of mortality and morbidity [1]. The excess glucose can be transformed to many potential precursors of glycation, such as glyoxal (GO), methylglyoxal (MG), and 3-deoxyglucosone (3-DG). Glycation is a spontaneous non-enzymatic reaction between reducing sugars or their reactive metabolites such as MG and protein, via rearrangement, oxidation, dehydration, and polymerization to generate advanced glycation-end products (AGEs). AGEs ultimately trigger several severe diabetic complications. such as cardiovascular diseases, nephropathy, neuropathy, and retinopathy [2]. Methylglyoxal (MG) is one of the reactive carbonyl species that plays an important role in glycation as a glycation precursor. Many recent studies have revealed that diabetic patients display levels of MG that are higher than normal people [3,4,5,6]. In addition, many reports indicate that MG is the most potent glycating agent, among the dicarbonyl compounds [7,8]. Aminoguanidine (AG) is a well-known anti-glycating agent that inhibited the formation of AGEs. Unfortunately, AG has been terminated in the clinical phase of medicine development, due to serious side effects such as congestive heart failure, myocardial infarction, gastrointestinal disturbance, and anemia [9]. Therefore, study on glycation inhibitors from nature has emerged as an intriguing approach to produce new candidates for the therapy of AGEs-associated diseases [10].

*Boesenbergia rotunda* Schltr. (*Syn*. *Boesenbergia pandurata*, *Kaempferia pandurata*), is a culinary herb of the Zingiberaceae family that is found in numerous Asian countries such as Indonesia, Malaysia, Thailand, India, and China. It is ordinarily utilized as a flavoring in food, such as curry, sauce, and soup, due to its aroma flavor, which advances craving. This herbal plant is also utilized as a conventional medicine to treat sicknesses such as muscle torment, rheumatism, gout, febrifuge, gastrointestinal disorders, flatulence, carminative, stomachache, dyspepsia, and peptic ulcer. The fresh rhizomes are utilized to treat inflammatory diseases, such as tooth decay, gum diseases, dry cough, cold, swelling, dermatitis, wounds, and diarrhea. According to Thai ethnomedicinal utilization, this plant is used as an aphrodisiac. Moreover, consumption of the leaves can relieve food allergies and poisoning. In Thailand, some acquired immunodeficiency syndrome (AIDs) patients used this plant as self-medication. In spite of the lack of scientific evidences to demonstrate the ethnomedicinal utilizations of this plant, the success of current biological research seems to clarify the significance of its traditional utilization. Several flavonoids were isolated and identified from the rhizome extract of *B. rotunda* such as flavones, flavanones, chalcones, and pimarane diterpenes [11,12]. However, scientific evidences and reports to the related inhibitory effects against DM and AGEs have not been documented so far.

In the present exploration, we applied an AGEs inhibition assay to screen Thai medicinal plants. We found a promising level of inhibition in the CH_2_Cl_2_ extract of *B. rotunda* rhizomes. An attempt to identify glycation inhibitors was made and reported herein for the first time.

## 2. Results and Discussion

### 2.1. Plant Isolation

The small air-dried pieces of fingerroot were submerged in hexane, dichloromethane, and methanol three times each, at room temperature, respectively. The dichloromethane extract was subjected to Sephadex LH-20 column chromatography, and eluted with MeOH to yield five fractions. The combined fractions 3 and 4 were subjected to silica gel column chromatography, and crystallized to yield three flavanones (pinocembrin; **1**, pinostrobin; **2**, and alpinetin; **3**), two chalcones (cardamomin; **4** and boesenbergin B; **5**), two dihydrochalcones (panduratin A; **6** and isopanduratin A; **7**), and one kavalactone (demethoxyyangonin; **8**). The structures of the isolated compounds (Figure 1) were elucidated on the basis of a detailed spectroscopic analysis, including ^1^H-NMR and ^13^C-NMR spectroscopy techniques.

### 2.2. Glycation Inhibition

#### 2.2.1. Effect of Isolated Compounds on Fluorescent-AGEs Formation Activity by an MG-BSA Assay

Glycation is a severe cause of diabetic complications by the accumulation of AGEs during our lifetime. Nevertheless, there are several studies that indicate that glycation inhibition contributed by synthetic agents such as aminoguanidine (AG) shows serious side effects. Thus, natural glycation inhibitors from edible plants are the alternative choices. This research started with the preliminary screening of compounds from many edible plants against MG-trapping activity. Of the compounds screened, pinocembrin from fingerroot revealed promising trapping activity. Therefore, fingerroot was selected for further study to search for other active components, in addition to pinocembrin (**1**).

All isolated compounds were evaluated for their inhibitory effect on the fluorescent-AGEs formation after incubation with bovine serum albumin (BSA) and MG for 24 h (Figure 2). Alpinetin (**3**) showed the highest inhibition (50%) against the MG-derived AGEs formation, followed by demethoxyyangonin (**8**, 46%), cardamomin (**4**, 44%), panduratin A (**6**, 43%), boesenbergin B (**5**, 39%), pinocembrin (**1**, 38%), isopanduratin A (**7**, 36%), and pinostrobin (**2**, 19%), respectively. Interestingly, most isolated compounds showed higher inhibition (36–50%) against the formation of MG-derived AGEs than anti-glycating agent aminoguanidine (AG, 28%), except for pinostrobin (**2**, 19%), as shown in Figure 2. This result indicated that the isolated compounds might be potent anti-glycation candidates. In addition, the results of AGEs inhibition observed in our work were comparable to those studied by Ma and coworkers [13] at the same concentration of AG (0.1 mM).

#### 2.2.2. Effect of Isolated Compounds on MG-Trapping Activity

As a result of the higher inhibition of isolated compounds against AGEs formation, MG-scavenging activity was subsequently examined, to elucidate whether the isolated compounds inhibited AGEs formation through the direct trapping of MG. Figure 3 demonstrates the MG-trapping activity of isolated compounds, whose percentages of trapping were reported as the value related to AG (100%). Of isolated compounds, pinocembrin (**1**) demonstrated the most potent trapping activity with the value of 109%, followed by panduratin A (**6**, 60%), cardamomin (**4**, 28%), demethoxyyangonin (**8**, 23%), boesenbergin B (**5**, 19%), alpinetin (**3**, 14%), isopanduratin A (**7**, 9%), and pinostrobin (**2**, 5%), respectively. It can be inferred that the trapping activity of **1** was comparable to that of AG. To determine whether the inhibitory effect of AGEs is contributed to by MG-trapping activity, Pearson’s correlation analysis (Table 1) was carried out. The small correlation coefficient (*r* = 0.159), together with *p* = 0.707, indicated a very weak positive correlation between the inhibitory effect of AGEs, and MG-trapping activity. In other words, the higher AGEs inhibitions of particular compounds such as alpinetin (**3**) over pinocembrin (**1**) were potentially derived from other effects.

As the small correlation coefficient from Pearson’s correlation, this analysis suggested that AGEs inhibition was not mainly contributed to by MG-trapping activity. It should be noted that AGEs inhibition of isolated flavonoids in this work could not be directly compared with those previously reported, due to different detection methodologies, such as fluorescent AGEs, non-fluorescent AGEs, glycated proteins, and radicals [2,4,14,15,16]. Moreover, AGEs formation inhibition is generated by many mechanisms, including MG-trapping activity. Therefore, the correlation could be interfered with by other mechanisms. For example, Sompong [14] and Hwang [17] found that there was no correlation between the percentage AGEs inhibition and radical scavenging activity, whereas Mutsuda [18] suggested that strong AGEs inhibition activity tended to exhibit strong radical scavenging activity. 

Because there are variations in core structures, together with types and positions of substitution groups, SAR of isolated compounds on MG-trapping was carefully investigated in six cases. Each case focused on the structural differences, as follows.
The presence of a methoxy group on the aromatic ring A of flavanone.The positions of methoxy groups on the aromatic ring A of flavanone.The ring C structure of flavanone and the α-β unsaturated ketone structure of chalcone.The presence of a geranyl group on chalcone.The positions of geranyl groups on ring A (chalcone), or at α-β unsaturated ketone (dihydrochalcone).The positions of the methoxy group on the aromatic ring A of dihydrochalcone.

Of flavanones (**1**–**3**), the presence of a methoxy group (OMe) on the aromatic ring A apparently decreased their potency against MG-trapping (Figure 4). Pinocembrin (**1**), whose structure contains no methoxy group, revealed MG-trapping activity (109%) that was comparable to that of AG (100%). On the other hands, flavanones **2** and **3** demonstrated very low trapping activity (5–14%). Although methoxy and hydroxy groups are both electron-donating groups and they enhance the electronic density to the aromatic ring, a greater steric hindrance of the methoxy group is likely to retard the aromatic ring A of flavanones **2** and **3** toward MG-trapping potency. However, the position of the methoxy group on ring A showed no significant difference in the trapping activities of **2** and **3**.

We further analyzed the lack of ring C in chalcones and dihydrochalcones, because they were dominant metabolites found in this plant. Direct comparison between alpinetin (**3**) and cardamomin (**4**) was present herein, since they are structural isomers and they contain similar substitution groups. Cardamomin (**4**) showed significantly higher trapping activity (28%) than alpinetin (**3**, 14%), thus indicating that the lack of ring C in chalcone did not reduce the trapping activity toward MG. Noticeably, the presence of α-β unsaturated ketone in **4** possibly alleviated the electron-withdrawing potency of a carbonyl group attached to ring A (Figure 5).

The presence and position of the geranyl groups of chalcones and dihydrochalcones toward trapping activity were also analyzed. Herein, compounds **4**, **5**, and **7** were directly compared, because they possessed identical patterns of hydroxy and methoxy groups on ring A. The geranyl group was likely to suppress trapping ability (Figure 6); however, significantly reduced activity was found in isopanduratin A (**7**). Although the presence of the geranyl group on ring A in boesenbergin B (**5**) was expected to produce steric hindrance, a decrease in trapping activity was not significantly observed.

The effect of the position of methoxy group of dihydrochalcone on the MG-trapping activity has been investigated by direct comparison between **6** and **7**. They are structural isomers that are different in the methoxy position; 4-OMe in **6**, and 6-OMe in **7**. The activity of compound **6** (60%) was significantly higher than that of compound **7** (9%) (*p* = 0.020 < α = 0.05) (Figure 7). In addition, the presence of hydrogen bonding between 6-OH and the carbonyl group at C-1 on ring A in **6** might promote the MG-trapping ability. This result was also observed in the previous reports [19,20].

Because pinocembrin (**1**) is the most potent glycation inhibitor, it was further evaluated for half the maximal effective concentration (EC_50_) of the MG-trapping activity. Pinocembrin (**1**) displayed a high efficiency against methylglyoxal, with an EC_50_ value of 63.22 ± 10.12 µM.

### 2.3. α-Glucosidase Inhibitory Activity and Kinetic Study of Pinocembrin (***1***)

For a diabetes patient, the reduction of glucose uptake is very important, which can be done through suppressing carbohydrate digestion. It is of interest to evaluate the α-glucosidase inhibition of **1** against rat intestinal maltase and sucrase, in the hope that it would possess multifunctions that are beneficial to controlling diabetes and its complications. This is the first time that pinocembrin (**1**) was tested for α-glucosidase inhibitory activity. Pinocembrin (**1**) displayed a moderate inhibition against both maltase and sucrose, with IC_50_ values of 0.35 ± 0.021 and 0.39 ± 0.020 mM, respectively (Appendix A).

In order to gain further insight into how pinocembrin (**1**) interacts with rat intestinal maltase and sucrase, the inhibition mode of pinocembrin (**1**) was analyzed by kinetic study. The Lineweaver–Burk plot of pinocembrin (**1**) against maltase (Figure 8a) displayed a series of straight lines. The intersections of all the straight lines were in the second quadrant. Kinetic examination showed that *V*_max_ decreased with elevated *K*_m_ in the presence of increasing concentrations of pinocembrin (**1**). This behavior recommended that pinocembrin (**1**) inhibited maltase in a mixed-type manner comprising two diverse pathways, competitive and non-competitive. The observed result was explained by simultaneous formation of enzyme–inhibitor (EI) and enzyme–substrate–inhibitor (ESI) complexes in competitive and non-competitive manners, respectively (Figure 9). We examined the pathway in which pinocembrin (**1**) was preferably preceded, by determining the dissociation constants of EI (*K*_i_) and ESI (*K*_i_′) complexes (Table 2). Clearly, the secondary plots (Figure 8b,c) illustrated *K*_i_ and *K*_i_′ values of 93 and 138 µM, respectively, thus indicating that pinocembrin (**1**) was overwhelmingly bound to maltase (EI), rather than the ESI complex formed. The putative inhibitory mechanism is summarized in Figure 9.

The inhibitory mechanism of pinocembrin (**1**) against sucrase (Figure 10) was also examined by utilizing the above strategy. Apparently, pinocembrin (**1**) inhibited sucrase through both competitive and non-competitive strategies (mixed-type inhibition). All kinetic variables are summarized in Table 2.

## 3. Material and Method

### 3.1. Plant Material and Isolation

Fingerroots (5 kg) were purchased from local shop number 43 in Sam-Yan Market, Phatumwan, Bangkok, Thailand, during January, 2015. The rhizomes were cleaned and cut into small pieces before air drying. The dried plant (1.1 kg) was submerged in hexane, dichloromethane, and methanol three time, each at room temperature, respectively. The dichloromethane extract was subjected to Sephadex LH-20 column chromatography, and eluted with MeOH, to yield five fractions (fractions 1–5). Fraction 3 was crystallized by MeOH to yield alpinetin (**3**, 150 mg) [21], while the remaining mother liquor was subjected to silica gel column chromatography (Merck Art 7730) and eluted with 25:10:65 CH_2_Cl_2_-MeOH-n-hexane to attain six fractions (fractions A-F). Fraction B was subjected to silica gel column chromatography, and eluted in a stepwise fashion with n-hexane, 1.5:8.5 CH_2_Cl_2_-n-hexane, 4:6 CH_2_Cl_2_-n-hexane, 8:2 CH_2_Cl_2_-n-hexane, CH_2_Cl_2_, 1:9 MeOH-CH_2_Cl_2_, and 2:8 MeOH-CH_2_Cl_2_ to yield six sub-fractions (sub-fractions B1-B6). Sub-fraction B3 was subjected to silica gel column chromatography, and eluted with 1:9 MeOH-CH_2_Cl_2_, yielding boesenbergin B (**5**, 80 mg) [22]. Fraction C was subjected to silica gel column chromatography and eluted with 1:49 EtOAc-n-hexane to yield pinostrobin (**2**, 200 mg) [21] and demethoxyyangonin (**8**, 70 mg) [23]. Fraction D was subjected to silica gel column chromatography, and eluted with 1:1 CH_2_Cl_2_-n-hexane to afford five sub-fractions (sub-fractions D1-D5). The combined sub-fractions D2 and D4 were crystallized by MeOH to yield isopanduratin A (**7**, 50 mg) [24] and panduratin A (**6**, 100 mg) [25], respectively. Fraction 4 was crystallized by MeOH to yield cardamomin (**4**, 150 mg) [21], while the remaining mother liquor was subjected to silica gel column chromatography and eluted with 1:49 EtOAc-n-hexane to yield pinocembrin (**1**, 180 mg) [21]. The structures of isolated compounds were elucidated on the basis of a detailed spectroscopic analysis, including ^1^H-NMR and ^13^C-NMR, and compared with previous reports.

### 3.2. In Vitro Glycation of Bovine Serum Albumin (BSA) by MG-BSA Assay

Glycated BSA was carried out according to our previous study [15]. The mixture of bovine serum albumin (BSA, 10 mg/mL), isolated compound (0.1 mM in dimethyl sulfoxide, DMSO), and methylglyoxal (MG, 0.25 mM) in 0.1 M phosphate buffer solution (PBS, pH 7.4) was incubated at 37 °C for 24 h. The glycated BSA was monitored by a spectrofluorometer (Fluroskan Ascent FL, Thermo Scientific, Barrington, IL, USA) at the excitation wavelength of 355 nm, and an emission wavelength of 460 nm. Aminoguanidine hydrochloride (AG) was utilized as a positive control. The percentage inhibition of MG-derived AGE formation was calculated by utilizing the formula below:(1)$$\mathrm{Inhibition}\;\mathrm{of}\;\mathrm{AGEs}\;\mathrm{formation}\;\left(\%\right)=\frac{\left(\mathrm{FC}-\mathrm{FCB}\right)-\left(\mathrm{FS}-\mathrm{FSB}\right)}{\mathrm{FC}-\mathrm{FCB}}\times 100$$
where FC and FCB are the fluorescent intensities of the control with and without MG, respectively. FS and FSB are the fluorescent intensities of the sample with and without MG, respectively.

### 3.3. Determination of Direct MG-Trapping Activity by HPLC

The ability of the isolated compounds to trap MG was carried out, according to our previous report [14]. The reaction mixture of MG (0.1 mM), isolated compound (0.1 mM in DMSO) and 0.1 M PBS (pH 7.4) was incubated at 37 °C for 24 h. The quantitative amount of the remaining MG was determined by derivatization with *o*-phenylenediamine (*o*-PDA) to convert to 2-methylquinoxaline (2-MQ). Then, 2-MQ obtained was measured by HPLC using 5-methylquinoxaline (5-MQ) as an internal standard. The mixture comprising 100 µL of *o*-PDA (20 mM) and 100 µL of 5-MQ (5 mM) was added into the sample vials, and incubated at 25 °C for 30 min. HPLC analysis was performed on a Shimadzu HPLC system (binary pump; model LC-10A, an auto-injector; model SIL-10A, and a UV detector; model SPD-10A) equipped with an Inertsil ODS-3 V C18 column (4.6 × 150 mm, 5 μm particle diameter). The column was flushed with a mixture of 1:1 methanol–water at a flow rate of 1 mL/min, and a 10 μL of injection volume. The overall running time was 15 min, and the wavelength used for detection was 315 nm. The area under each chromatogram peak, 2-MQ and 5-MQ, was calculated with an LC solution program. Peak integrality proportions of 2-MQ to 5-MQ were utilized for quantitative examination. The amount of MG was calculated by utilizing the standard curve of 2-MQ/5-MQ proportion. AG was utilized as a positive control. The percentage MG-trapping was calculated utilizing the formula below:(2)$$\%\mathrm{MG}-\mathrm{trapping}=100-\left[\frac{\mathrm{Amount}\;\mathrm{of}\;\left(\mathrm{MG}\;\mathrm{in}\;\mathrm{sample}-\mathrm{MG}\;\mathrm{in}\;\mathrm{positive}\;\mathrm{control}\right)}{\mathrm{Amount}\;\mathrm{of}\;\mathrm{MG}\;\mathrm{in}\;\mathrm{positive}\;\mathrm{control}}\times 100\right]$$

### 3.4. α-Glucosidase Inhibitory Activity

The α-glucosidase (from rat intestine) inhibitory effect was evaluated, utilizing our previous study [26]. Rat intestinal maltase and sucrase, a crude enzyme solution, was prepared from rat intestinal acetone powder. A solution of rat intestinal acetone powder (1 g) in 0.9% NaCl solution (30 mL) was centrifuged (12,000 rpm) for 30 min, and then the aliquot obtained was used for the assay. The mixture of 10 μL of isolated compounds (1 mg/mL in DMSO), 30 μL of the phosphate buffer (0.1 M, pH 6.9), and 20 μL of the substrate solution (10 mM for maltose and 100 mM for sucrose in phosphate buffer) was added to a glucose assay kit (80 μL) and the crude enzyme solution (20 μL), which then was incubated at 37 °C for 10 min and 40 min for maltose and sucrose, respectively. The enzymatic activity was measured at 503 nm by using a microplate reader, model 3550 UV. The percentage inhibition was calculated by [(A_c_ − A_s_)/A_c_] × 100, where A_c_ is the absorbance of the control, and A_s_ is the absorbance in the presence of sample. The results were reported as the IC_50_ value, and the experiment was carried out in triplicate. Acarbose^®^ was used as a standard control.

### 3.5. Kinetic Study of α-Glucosidase Inhibition

The kinetic analysis of pinocembrin was determined by using an increasing concentration of substrates maltose (2–10 mM) and sucrose (20–100 mM), and a Lineweaver–Burk plot was performed to determine the type of inhibition. The secondary plot was utilized to determine the *K*_i_ and *K*_i_′ values, which were replotted between the slope and intercept from the Lineweaver–Burk plot vs the various concentrations of pinocembrin.

### 3.6. Statistical Analysis

The data are presented as means ± standard error of mean (SEM; *n* = 3). In the experiment for MG trapping activity, a paired sample *t*-test was evaluated for significant differences, as shown by the stars above the graphs, between each pair of samples. For the significant differences that are shown as the letters above the graphs, among compounds, one-way ANOVA (Analysis Of Variance) was evaluated, and Duncan’s post hoc test was used for mean comparisons. Pearson’s correlation analysis was used to determine the correlation between AGEs formation inhibition activity and MG-trapping activity. A *p*-value < 0.05 was considered to be statistically significant.

## 4. Conclusions

Three flavanones (pinocembrin, pinostrobin, and alpinetin), two chalcones (cardamomin and boesenbergin B), two dihydrochalcones (panduratin A and isopanduratin A), and one kavalactone (demethoxyyangonin) were isolated from a dichloromethane extract of fingerroots. Most of isolated compounds showed higher levels of inhibition against the formation of MG-derived AGEs than the anti-glycating agent, aminoguanidine (AG, 28%). Moreover, studies of structure–activity relationship (SAR) were investigated in this research by using comparisons of the structures of flavonoids with their MG-trapping activities. The structural requirements of flavonoids on this activity were obtained, and the results showed the following: (1) a hydroxy group and an α-β unsaturated ketone structure could facilitate the activity; (2) methoxy and geranyl groups could reduce the activity; and (3) a methoxy group at the C-4 position of dihydrochacone is more active than at the C-6 position. The small Pearson’s correlation coefficient (*r* = 0.159) indicated a very weak positive correlation between the AGEs inhibitory effect and the MG-trapping activity. Of the compounds examined, pinocembrin (**1**) demonstrated the highest trapping activity, with a value of 109%. It can be inferred that the trapping activity of **1** was comparable to that of AG, and that it displayed a high efficiency against methylglyoxal, with an EC_50_ value of 63.22 ± 10.12 µM. In addition, this is the first time that pinocembrin was tested for α-glucosidase inhibitory activity. Pinocembrin displayed a moderate level of inhibition against both maltase and sucrose, with IC_50_ values of 0.35 ± 0.021 and 0.39 ± 0.020 mM, respectively. The highly potent level of glycation (MG-trapping activity) and α-glucosidase, and inhibitions of pinocembrin (**1**) could be beneficial in diabetes treatment, as well as in preventing the onset of its complications.

## Figures and Tables

**Figure 1 molecules-23-03365-f001:**
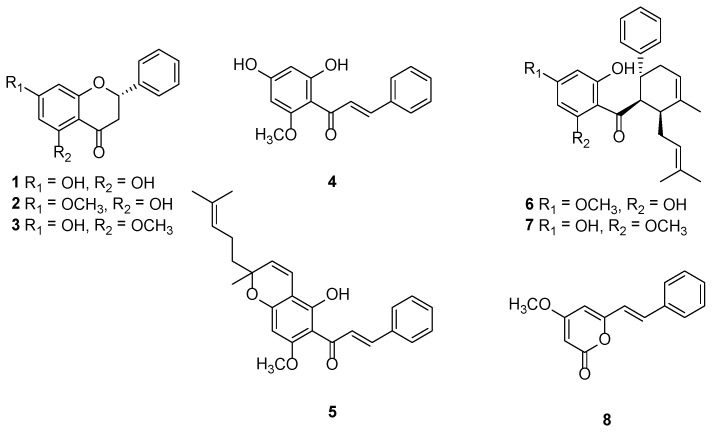
Structures of the isolated compounds (**1**–**8**).

**Figure 2 molecules-23-03365-f002:**
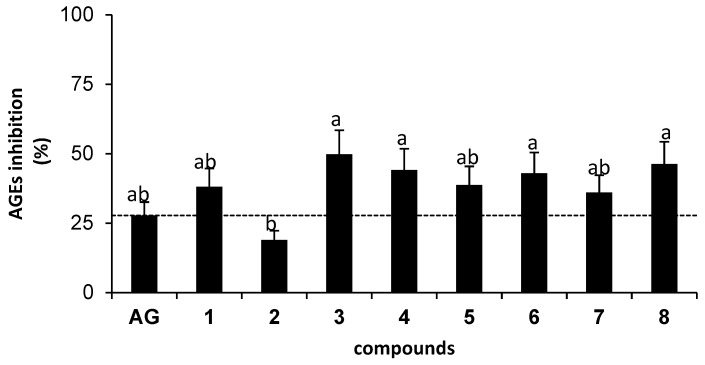
Advanced glycation-end products (AGEs) inhibition activity of aminoguanidine (0.1 mM) and isolated compounds (0.1 mM) on MG-derived AGEs formation inhibition in BSA. The results are presented as mean ± SEM for *n* = 3. Different letters show significant differences analyzed by ANOVA with Duncan’s post hoc test (*p* < 0.05).

**Figure 3 molecules-23-03365-f003:**
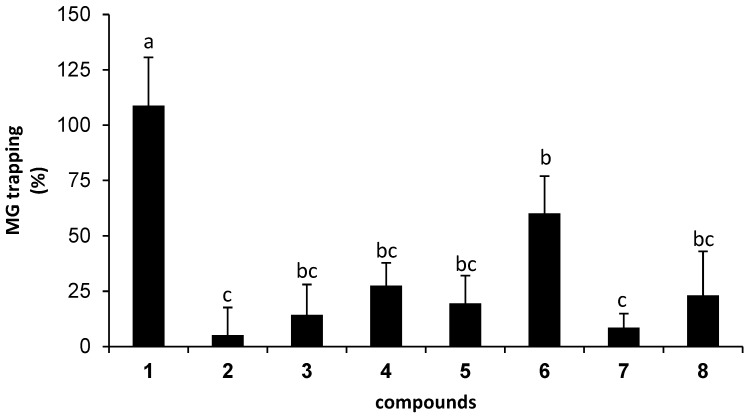
The percentage of MG-trapping abilities of isolated compounds (0.1 mM). The results are presented as the mean ± SEM for *n* = 3. Different letters show significant differences, as analyzed by ANOVA, with Duncan’s post hoc test (*p* < 0.05).

**Figure 4 molecules-23-03365-f004:**
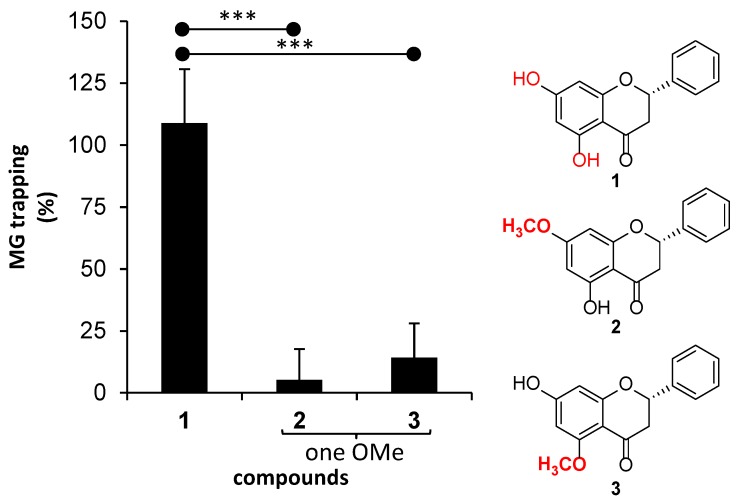
The effect of the presence of one methoxy group substitution on the aromatic ring A of flavanone on MG-trapping activity. The results are presented as the mean ± SEM for *n* = 3. The stars (***) showed significant differences (*p* < 0.001) between **1** and **2**, and between **1** and **3**, using a paired sample *t*-test.

**Figure 5 molecules-23-03365-f005:**
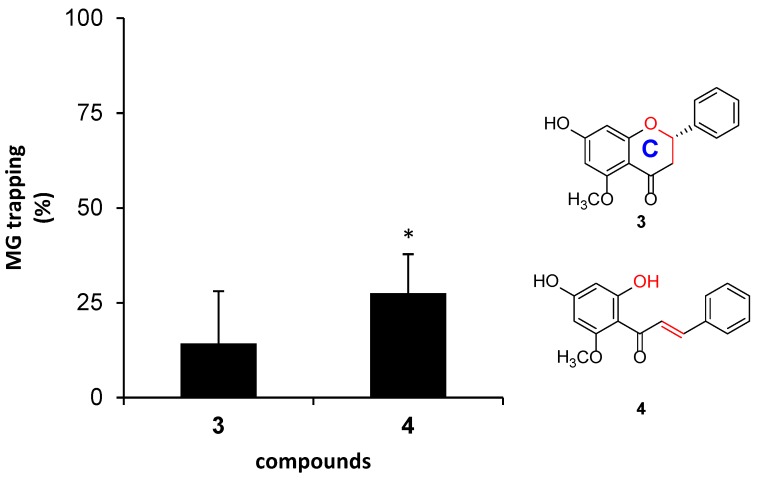
The effect of ring C on the structure of the flavanone, and the lack of the ring C on the structure of chalcone against MG. The results are presented as mean ± SEM for *n* = 3. The star (*) showed a significant difference (*p* < 0.05) between **3** and **4** using paired sample *t*-test.

**Figure 6 molecules-23-03365-f006:**
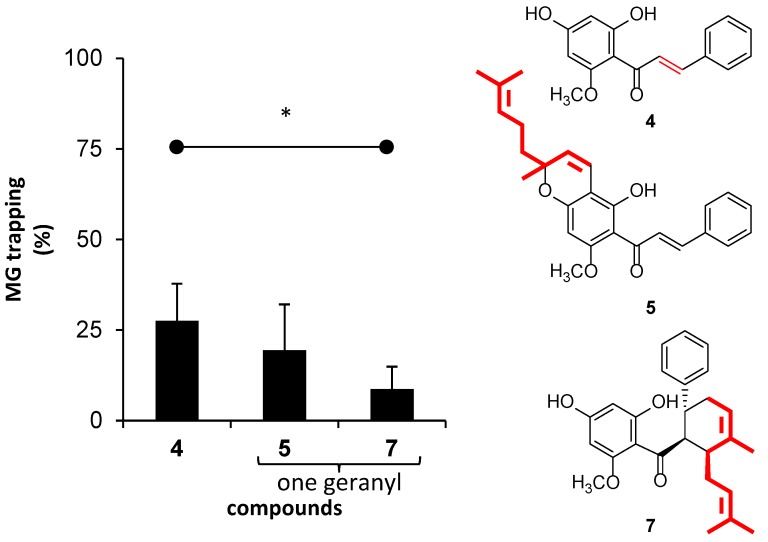
The effect of the presence of one geranyl group on the structure of chalcone on MG-trapping activity. The results are presented as mean ± SEM for *n* = 3. The star (*) showed significant differences (*p* < 0.05) between **4** and **7** when using a paired sample *t*-test.

**Figure 7 molecules-23-03365-f007:**
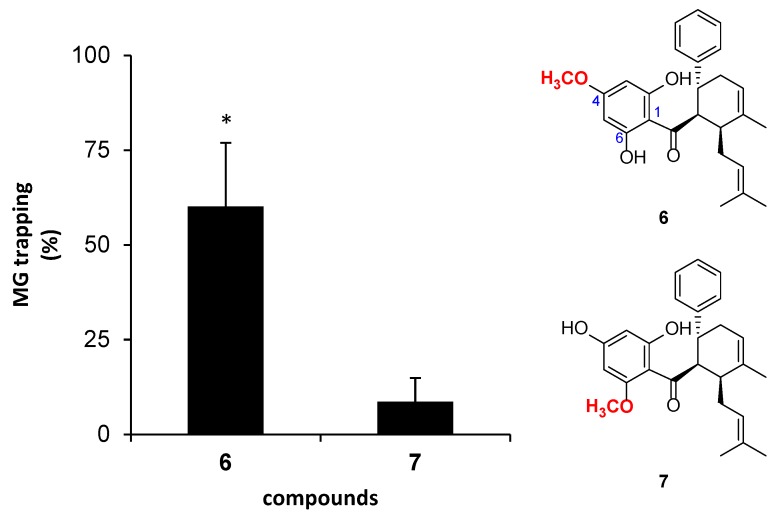
The effect of the position of methoxy group substitution on the aromatic ring A of dihydrochalcone on MG-trapping activity. The results are presented as mean ± SEM for *n* = 3. The star (*) showed significant difference (*p* < 0.05) between **6** and **7**, using a paired sample *t*-test.

**Figure 8 molecules-23-03365-f008:**
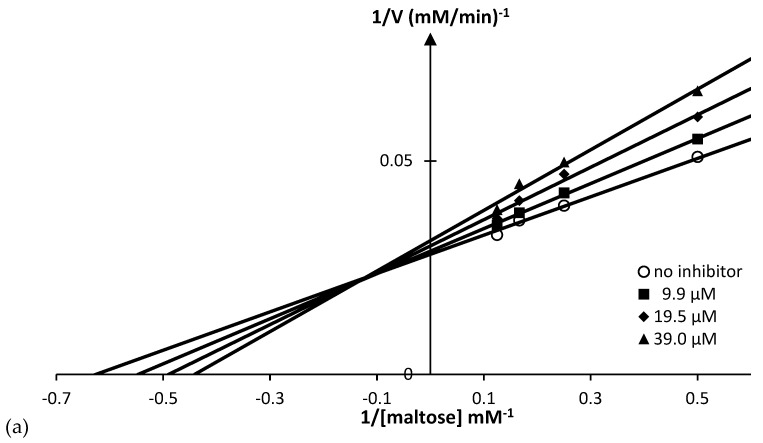
Kinetic study plots of pinocembrin (**1**) against rat intestinal maltase; (**a**) Lineweaver–Burk plot; (**b**) secondary plot of the slope and concentration of pinocembrin (**1**) for the determination of *K*_i_; (**c**) secondary plot of intercept and concentration of pinocembrin (**1**) for the determination of *K*_i_′.

**Figure 9 molecules-23-03365-f009:**
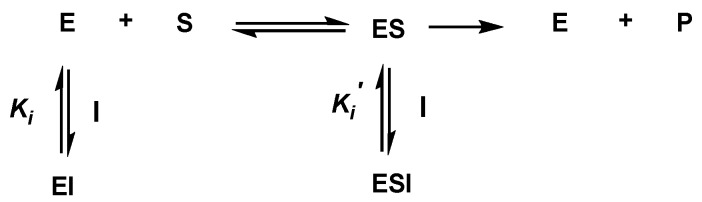
Putative inhibitory mechanism of pinocembrin (**1**) against rat intestinal α-glucosidases. E, S, I, and P represent enzyme, substrates (maltose and sucrose), inhibitor (pinocembrin), and glucose, respectively.

**Figure 10 molecules-23-03365-f010:**
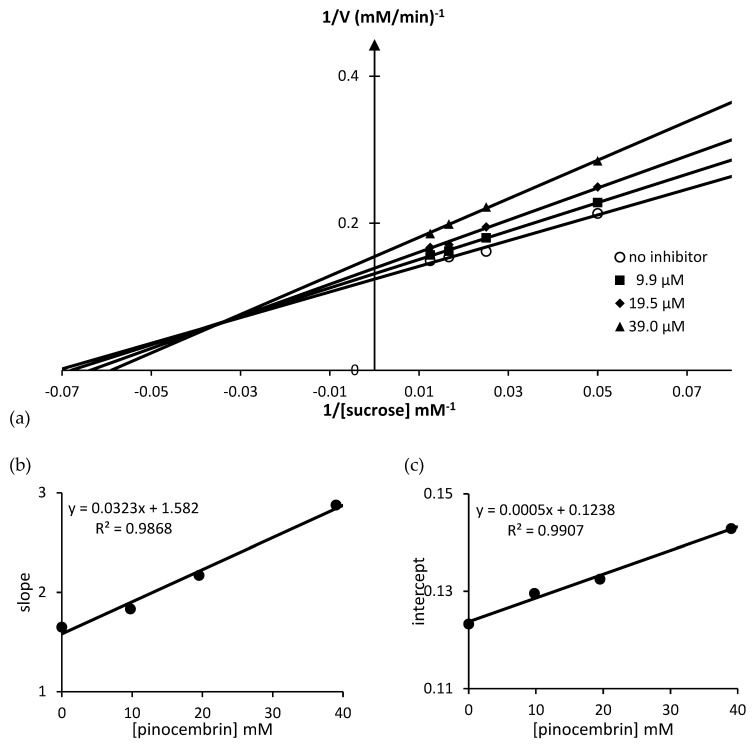
Kinetic study plots of pinocembrin against rat intestinal sucrase; (**a**) Lineweaver–Burk plot; (**b**) secondary plot of the slope and concentration of pinocembrin for the determination of *K*_i_; (**c**) secondary plot of the intercept and concentration of pinocembrin for the determination of *K*_i_′.

**Table 1 molecules-23-03365-t001:** Pearson’s correlation analysis of AGEs formation inhibition activity and the MG-trapping activity of isolated compounds.

Activity	MG-Trapping	
	Correlation Coefficient	*p*-Value (2-Tailed)
AGEs formation inhibition	0.159	0.707

**Table 2 molecules-23-03365-t002:** Kinetic factors of pinocembrin (**1**) for rat intestinal α-glucosidase inhibition.

Enzyme	*K*_i_ (µM)	*K*_i_′ (µM)	Inhibition Type
Maltase	93	138	Mixed
Sucrase	51	253	Mixed

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
