# Peer review of "Identification of Pinocembrin as an Anti-Glycation Agent and α-Glucosidase Inhibitor from Fingerroot (*Boesenbergia rotunda*): The Tentative Structure–Activity Relationship towards MG-Trapping Activity"

_molecules, 2018, doi:10.3390/molecules23123365_

Reviewer 1 Report

The manuscript is dealing with the  identification of pinocembrin as anti-glycation and  anti-diabetic agent from fingerroot (Boesenbergia rotunda). Beside this, the tentative structure-activity relationship towards MG-trapping activity was evaluated.

The work is nice but sill needs to be improved before publication.

-Check English

-the scheme presented at section 2.1 is not usefull because all the information are well described in section 3.1

- conclusions should be more specific including concret practical applications.

Author Response

Response to Reviewer 1 Comments

1. Check English

-The revised manuscript was carefully checked for English by senior professor in our group due to time limit to submit by 16 Dec. We promise to send the revised manuscript to Native English speaker (at University Language Institute) for checking English; however, it would be the last step.

2. The scheme presented at section 2.1 is not useful because all the information are well described in section 3.1

-Figure 1 was deleted from the manuscript.

3. Conclusions should be more specific including concret practical applications.

-We make additional conclusion on practical applications.

Reviewer 2 Report

The manuscript is written by Potipiranun et al., is to examine the anti-glycation effects of isolated pinocembrin from fingerroot. The results indicated AGEs and α-glucosidase inhibition activity and the tentative SARs towards MG-trapping activity. Results of this study would be very interesting to the readers in the area of natural product and medicinal chemists. My comments are as follows: 

Title:  Title should be changed. This is only α-glucosidase inhibition assay. There are so many targets for anti-diabetic activity.

Abstract: line 28-29. should be removed. 

              the value of α-glucosidase inhibition by pinocembrin should be addressed. 

              MG-trapping potency was relatively comparable to AG (But MG-trapping potency of AG is               missing in the results).

Materials and Methods: 

1. To allow other scientists to get the reproducible results, the author may purchase the Fingerroots from the authentic company or should be identified by taxonomic scientist and the identification no. must be included in the MS. 

2. since all the compounds identified from dichloromethane extract. so why authors are using hexane and methanol extract. rational this statement.

3. Compounds 1-3, the C-NMR data are missing. and also which references authors are using  to compare the NMR data for structure elucidation. please addressed. 

4. what is the control to check the AGEs inhibition assay. 

5. MG-trapping ability by HPLC: Methods should be written more details. For example: what is the meaning of 2-MQ? what is the control mixture? what is the positive control used for comparison?

6. Since compounds 3-8 have a very good inhibition activity towards AGEs formation. it is suggested to measure the α-glucosidase inhibition activity of all isolated compounds and give it as supplementary data.

Results: 

1. Figure 3: statistical analysis is absent. should be addressed. 

2. Figure 4: positive control is missing. how authors calculate the statistical significance where control data is missing?

3. Authors should include some HPLC figure of MG-trapping assay. For example, 2-MQ, 5-MQ band intensity for control, positive control and with the presence of pinocembrin.   

Discussion 

1. Need more scientific comparison of AGEs formation inhibition activity of various flavonoids. Authors should be cited in the following article ( reference 1-5).  

2. Paragraph 3 also need more scientific discussion and cross-reference of flavonoids SAR study. 3. paragraph 4, Authors claimed that pinocembrin had higher activity (9%) than positive control AG. but the results in missing. please addressed this

1.      Hwang SHWang ZGuillen Quispe YNLim SSYu JM. Evaluation of Aldose Reductase, Protein Glycation, and Antioxidant Inhibitory Activities of Bioactive Flavonoids in Matricaria recutita L. and Their Structure-Activity Relationship. J Diabetes Res. 2018, 2018:3276162.

2.      Li X, Liu GJ, Zhang W, Zhou YL, Ling TJ, Wan XC, Bao GH. Novel Flavoalkaloids from White Tea with Inhibitory Activity against the Formation of Advanced Glycation End Products. J Agric Food Chem. 2018;66(18):4621-4629.

3.      Yeh WJ, Hsia SM, Lee WH, Wu CH. Polyphenols with antiglycation activity and mechanisms of action: A review of recent findings. J Food Drug Anal. 2017; 25(1):84-92.

4.      Chi-Hao Wu, and Gow-Chin Yen. Inhibitory Effect of Naturally Occurring Flavonoids on the Formation of Advanced Glycation Endproducts. J. Agric. Food Chem., 2005, 53 (8), pp 3167–3173. Yixi Xie and Xiaoqing Chen. Structures Required of Polyphenols for Inhibiting Advanced Glycation end Products Formation. Current Drug Metabolism, 2013, 14, 414-431 

Author Response

Response to Reviewer 2 Comments

Title:

1.1. Title should be changed. This is only α-glucosidase inhibition assay. There are so many targets for anti-diabetic activity.

-The title was changed to “Identification of pinocembrin as anti-glycation agent and -glucosidase inhibitor from fingerroot (Boesenbergia rotunda): The tentative structure-activity relationship towards MG-trapping activity.” as reviewer’s suggestion.

Abstract:     

2.1. Line 28-29 should be removed.

-We already removed as reviewer’s suggestion.

2.2. The value of α-glucosidase inhibition by pinocembrin should be addressed.

-We added the IC50 value of pinocembrin into abstract.

2.3. MG-trapping potency was relatively comparable to AG (But MG-trapping potency of AG is missing in the results).

-In this experiment, %MG-trapping of aminoguanidine (AG) was set as 100% trapping at the tested concentration (0.1 mM). Therefore, %MG-trapping of isolated compounds was shown as relative values, compared to aminoguanidine (AG). In our work, pinocembrin is the only compound that is more potent than AG.

Materials and Methods:           

3.1. To allow other scientists to get the reproducible results, the author may purchase the Fingerroots from the authentic company or should be identified by taxonomic scientist and the identification no. must be included in the MS.

-We added the information in detail of purchase place in this section.

3.2. Since all the compounds identified from dichloromethane extract. so why authors are using hexane and methanol extract. rational this statement.

-Actually, after we got three crude extracts (hexane, DCM, and MeOH extracts), we firstly checked the TLC of three crude extracts. We found that only DCM extract showed the interesting TLC profile. Therefore, we took only DCM to further isolate and purify. 

3.3. Compounds 1-3, the C-NMR data are missing. and also which references authors are using  to compare the NMR data for structure elucidation. please addressed.

-We first run the 1H NMR, and it was likely that compounds 1-3 are flavanones. Therefore, we confirmed their identity with the authentic flavanones (pinocembrin, pinostrobin and alpinetin) deposited in our laboratory. However, we also added the references containing NMR data of all isolated compounds in section 3.1.

3.4. what is the control to check the AGEs inhibition assay.

-Aminoguanidine (AG) is the positive control that we already put in the text.

3.5. MG-trapping ability by HPLC: Methods should be written more details. For example: what is the meaning of 2-MQ? what is the control mixture? what is the positive control used for comparison?

-We added more detail: 2-methylquinoxaline (2-MQ), and aminoguanidine (AG) is the positive control

3.6. Since compounds 3-8 have a very good inhibition activity towards AGEs formation. It is suggested to measure the α-glucosidase inhibition activity of all isolated compounds and give it as supplementary data.

-In this study, we focus on compound 1 because it showed most potent trapping ageist MG than positive control (AG). However, the mechanism on AGEs inhibition together with α-glucosidase inhibition of compounds 3-8 are currently studied, and the results will be published elsewhere.

Results:

4.1. Figure 3: statistical analysis is absent should be addressed.

-We added the statistical analysis of AGEs formation inhibition activity to this Figure.

4.2. Figure 4: positive control is missing. how authors calculate the statistical significance where control data is missing?

-In this experiment, %MG-trapping of aminoguanidine (AG) was set as 100% trapping at the tested concentration (0.1 mM). Therefore, %MG-trapping of isolated compounds was shown as relative values, compared to aminoguanidine (AG). In our work, pinocembrin is the only compound that is more potent than AG.

4.3. Authors should include some HPLC figure of MG-trapping assay. For example, 2-MQ, 5-MQ band intensity for control, positive control and with the presence of pinocembrin.

-We added HPLC chromatogram of MG standards, positive control and pinocembrin in supplementary material.

Discussion  

5.1. Need more scientific comparison of AGEs formation inhibition activity of various flavonoids. Authors should be cited in the following article ( reference 1-5). 

1.      Hwang SH, Wang Z, Guillen Quispe YN, Lim SS, Yu JM. Evaluation of Aldose Reductase, Protein Glycation, and Antioxidant Inhibitory Activities of Bioactive Flavonoids in Matricaria recutita L. and Their Structure-Activity Relationship. J Diabetes Res. 2018, 2018:3276162.

2.      Li X, Liu GJ, Zhang W, Zhou YL, Ling TJ, Wan XC, Bao GH. Novel Flavoalkaloids from White Tea with Inhibitory Activity against the Formation of Advanced Glycation End Products. J Agric Food Chem. 2018;66(18):4621-4629.

3.      Yeh WJ, Hsia SM, Lee WH, Wu CH. Polyphenols with antiglycation activity and mechanisms of action: A review of recent findings. J Food Drug Anal. 2017; 25(1):84-92.

4.      Chi-Hao Wu, and Gow-Chin Yen. Inhibitory Effect of Naturally Occurring Flavonoids on the Formation of Advanced Glycation Endproducts. J. Agric. Food Chem., 2005, 53 (8), pp 3167–3173.

5. Yixi Xie and Xiaoqing Chen. Structures Required of Polyphenols for Inhibiting Advanced Glycation end Products Formation. Current Drug Metabolism, 2013, 14, 414-431

-We added discussion and the above references suggested by reviewer in the lines between 140-150.

5.2. Paragraph 3 also need more scientific discussion and cross-reference of flavonoids SAR study.

- We added discussion in the lines between 140-150.

5.3. Paragraph 4, Authors claimed that pinocembrin had higher activity (9%) than positive control AG. but the results in missing. please addressed this

-Positive control (aminoguanidine: AG) is 100% inhibition.

Round  2

Reviewer 1 Report

The manuscript was improved so I recommend to be published.

Reviewer 2 Report

Thank you very much for your answer.